# Does Intense Endurance Workout Have an Impact on Serum Levels of Sex Hormones in Males?

**DOI:** 10.3390/biology12040531

**Published:** 2023-03-31

**Authors:** Michał Wiciński, Oskar Kuźmiński, Artur Kujawa, Witold Słomko, Anna Fajkiel-Madajczyk, Maciej Słupski, Artur Jóźwik, Karol Kubiak, Stephan Walter Otto, Bartosz Malinowski

**Affiliations:** 1Department of Pharmacology and Therapeutics, Faculty of Medicine, Collegium Medicum in Bydgoszcz, Nicolaus Copernicus University, M. Curie 9, 85-090 Bydgoszcz, Poland; 2Department of Physioterapy, Collegium Medicum in Bydgoszcz, Nicolaus Copernicus University in Torun, 87-100 Toruń, Poland; 3Department of Hepatobiliary and General Surgery, Faculty of Medicine, Collegium Medicum in Bydgoszcz, Nicolaus Copernicus University, M. Curie 9, 85-090 Bydgoszcz, Poland; 4Institute of Genetics and Animal Biotechnology, Polish Academy of Sciences, 05-552 Jastrzębiec, Poland; 5Department of Obstetrics and Gynecology, St. Franziskus-Hospital, 48145 Münster, Germany; 6Department of Urology, Raphaels Clinic, 48143 Münster, Germany

**Keywords:** testosterone, endurance training, estradiol, male, nitric oxide, SHBG

## Abstract

**Simple Summary:**

In this article, authors studied serum level changes of a few sex hormones among athletes. It is widely believed that endurance workout is related to an increase of serum levels of sex hormones in males, but our research proved otherwise.

**Abstract:**

The benefits of physical activity and sports are widely known and proved to be crucial for overall health and well-being. In this research, the authors decided to measure the impact of endurance training in a professional male rowing team on the serum concentration levels of testosterone, estradiol, sex hormone binding globulin (SHBG) and nitric oxide (NO) and apolipoprotein A1 (Apo-A1). Proper levels of the serum concentration are necessary in order to maintain physical effectiveness. Authors analyzed the data and reviewed the former conterminous articles to find the possible mechanisms leading to changes of serum concentration of certain hormones and molecules. The direct effect of physical activity was a decrease in testosterone serum concentration (from 7.12 ± 0.4 to 6.59 ± 0.35 (ng/mL)), sex hormone binding globulin serum concentration (from 39.50 ± 2.48 to 34.27 ± 2.33 (nmol/L)), nitric oxide serum concentration (from 440.21 ± 88.64 to 432 ± 91.89 (ng/mL)), increase in estradiol serum concentration (from 78.2 ± 11.21 to 83.01 ± 13.21 (pg/mL)) and no significant increase in Apo-A1 serum concentration (from 2.63 ± 0.2 to 2.69 ± 0.21 (mg/mL)). Low testosterone concentration in OTS may be a consequence of increased conversion to estradiol, because gonadotropic stimulation is maintained. Apo-A1 serum concentration was measured due to a strong connection with testosterone level and its possible impact of decreasing cardiovascular risk.

## 1. Introduction

Endurance training is proved to have an impact on serum levels of hormones in athletes. A group of male athletes were assessed in this research to find a relationship between the intensity of endurance training and serum levels of sex hormones. While it is widely believed that time and intensity of training sessions in training season linearly correlate with sex hormones, the following research proves otherwise.

An androgen, or male sex hormone, is defined as a substance capable of the induction of the well-known secondary sex characteristics associated with puberty: growth spurt, increased libido, increased erectile function, acne, excess body hair, increased muscle mass, deepening of the voice, spermatogenesis, gynecomastia (usually transient) [1]. Testosterone is the most common human hormone associated with muscle mass gain, physical strength and endurance. It also plays a major role in the development of the male reproductive system, as well as the process of secondary sexual characteristics promotion [2]. Modern athletes rely on the importance of testosterone in a similar fashion as occurring in the past. High serum levels of testosterone are beneficial in many sports that require strength, endurance and overall enhancement of one’s performance. Lower concentrations of testosterone in males are related to weaker bone mass, lethargy and dysfunctions of the reproductive system [3]. The range of testosterone levels in healthy males is determined between 264 and 916 (ng/mL) in the European and American population [4].

Sex hormone binding globulin (SHBG) is a glycoprotein which, due to its molecular structure, controls the bio-availability of other hormones. SHBG tends to bind other hormones and limit their presence in the bloodstream. It is primarily produced in the liver, but its origin is proven to exist in the hypothalamus and the pituitary gland in close relation to oxytocin-producing neurons [5]. During one’s lifetime, it is observed that a decrease in serum concentration of SHBG allows the bio-availability of the sex hormones both in males and females especially during the beginning of puberty. The serum concentration of SHBG is lowered by an increase in concentration of insulin, growth-like hormone and insulin-like growth factor 1 (IGF-1), whereas higher levels of estrogen and thyroxine tend to increase the serum concentration levels of SHBG. There are no standardized ranges of SHBG serum concentration. One research found that the difference between the lowest and highest values in one cohort can reach nearly 20-fold difference. This leads to the conclusion that the SHBG serum concentration cannot be examined as a singular marker but as a part of the hormonal loop, and the routine testing of this hormone should be performed with other hormones correlated to clinical indications [6,7].

Estradiol is a hormone related to the development of the female reproductive system as well as secondary sex characteristics. While the importance of the hormone is widely discussed due to its importance in clinical situations in females, it also plays a crucial role in maintaining the balance of the male hormonal system. Exposure of fetal testes to substances being able to mimic estrogens such as estradiol is proved to be a cause of decline in sperm count in later life [8]. The range of estradiol in healthy, fertile males is determined between 10 and 82 (pg/mL) [9].

Nitric oxide (NO) is a gas that is naturally produced in the human body from L-arginine by nitric oxide synthase enzymes. It was previously known as endothelial derived relaxing factor, as the primary place of the synthesis was already targeted, but the exact molecule was still unknown [10]. Being a free radical molecule, it is highly reactive within cells and diffuses across membranes. The main clinical significance of NO is its ability to dilate the blood vessels and therefore increase blood flow.

Apolipoprotein A-I (Apo A-I) is a constitutive component and a major structural and functional protein of high-density lipoprotein (HDL). It composes approximately 70% of HDL. Apo A-I plays an indispensable role in transporting excess cholesterol from peripheral cells to the liver and cellular cholesterol homeostasis. Its potential protection of the cardiovascular system and lowering cardiovascular disease risk with collaterally HDL enhancement result from its multifunctional role in immunity, inflammation, apoptosis and various types of infection [10].

The aim of this study was to investigate the relationship between endurance training and concentrations of testosterone, SHBG, estradiol, NO and Apo-A1 to suggest potential mechanisms for changes in sex hormones after endurance training/workout.

### Correlation between Level of Testosterone and Physical Efficiency

Testosterone retains nitrogen and is requisite in growth, development and gaining muscle mass. Studies showed that lower levels of androgen hormones effectively caused a decrease in lean muscle mass, strength and size. With the role that sex hormones play in maintenance and growth of bones, which is decreasing bone resorption and increasing bone mineral density through its aromatization to estradiol, it is indisputable that maintaining optimal levels of these hormones are essential to whole physical health for professional athletes [1,11,12].

Several studies have shown that there is an undeniable interaction between testosterone concentration and physical abilities. With higher levels of the hormone, athletes benefit from increased muscle size and strength, aerobic endurance, decreased fat mass, faster recovery from high exertion exercise and increased muscular power [13,14]. Different studies, on the contrary, indicated that athletes, who performed endurance exercise training resulted in significant diminution in resting testosterone, with some meeting the clinical criteria for androgen deficiency classification during the specified training regimen. Nevertheless, none of those athletes presented any physical performance deterioration. In fact, the opposite occurred and efficiency improved significantly and each respective performance test was also improved from the prior performances. In conclusion, the studies suggest that acute decrease in testosterone may not be entirely indicative of compromised physical efficiency whereas elements of the reproductive system and bone health may still be a potential menace [15].

## 2. Materials and Methods

Authors collected 47 samples for examination of the level of specific human hormones (testosterone, estradiol, SHBG, nitric oxide and Apo-A1). The participants were divided into two groups: control group (*n* = 24) and research group (*n* = 23, healthy adult males, professional athletes). Measurements were performed in two time points (time point 0: T_0_—before regular training and time point 1: T_1_—after 6 months of training season). The condition of blood sampling was identical in both T_0_ (before the training season) and T_1_ (after the training season)—the participants were asked to remain in a fasted state (with the last meal scheduled to be in the evening the day before the blood collecting). The samples were drawn between 9 AM and 10 AM in both T_0_ and T_1_ in a laboratory environment.

The ethics committee of Collegium Medicum in Bydgoszcz, Nicolaus Copernicus University, in Toruń approved the study protocol and design. Written, informed consent form were obtained from all the participants before the beginning of the study. Each test was conducted in concordance with the criteria set by the Declaration of Helsinki.

There are three main seasons in professional rowing. The first is the preparation season which lasts approximately six months from December to May. It is primarily focused on developing overall physical endurance and mental attitude towards competition. The second is the starting season which lasts five months (from the end of preparation season in May to October). It consists of higher intensity training specifically focused on rowing techniques—both tactics-wise and technique needed to develop peak condition in the rower. It is also crucial for maintaining strength in rowing competitors as well as developing endurance and speed. The third one is the transition season which lasts one month—November. Its main focus is moving from specialty rowing training into activities primarily focused on maintaining overall physical activity at an adequate level. This season is also best for healing injuries and overload of skeletal and muscle systems in participants.

One microcycle of training lasted for seven days in preparation and starting seasons.

The starting season consisted mainly of endurance training, strength training and high intensity training. Participants’ range of rowing was between 50 and 70 km in a week.

Aerobic training was focused on improving special endurance on a boat (4 days in a week for 50 min each) and on overall endurance mainly developed in cycling, running and swimming sessions (3 days in a week for 60 min each). Participants trained with heart rate (HR) reaching approximately 70–75% of maximum heart rate (HR max).

The strength training was divided into two training days for 75 min of training per day.

The high intensity training (with intensity of training reaching the lactate threshold of each participant) was performed three times during a season, each training for 30 min and with values of HR between 80–85%.

The microcycle of training in transition season was focused on aerobic activity. Participants rowed for 25 km per week. During this season, the main goal was to develop special endurance with two 60-min training sessions on a boat per week. This was performed in addition of one training on a rowing machine divided between two 45-min sessions once per week with HR values of 80% (based on individual predispositions of a participant). The overall endurance was developed for roughly 200 min per week through swimming, cycling or running sessions (with addition of swimming pool training 40 min twice a week and running sessions 60 min twice a week—HR values reaching 65–70% of HR max). Strength training was divided between one heavy session per week lasting for 60 min and one 20-min session per week as a strength training supplement.

### 2.1. Inclusion and Exclusion Criteria

Since the main aim of the study was to analyze a group of young adult males regularly involved in endurance training sessions, the most important inclusion criteria was membership in a rowing club.

Inclusion criteria for the study were volunteers aged between 18–25 years old, with active participation of at least 3 years of training seasons, which means every participant had a previous experience in endurance training. The main exclusion criteria were usage of any substances which may be regarded as an illegal enhancement of one’s physical abilities, however, none of the participants had a history of usage of such substances. The exclusion criteria also contained usage of any medications and body mass index above 25 kg/m^2^, as the aim of the study was to analyze healthy young males. None of the participants had a previous medical history of chronic disease and remained in such a state for the time of the study. Two of the members had a significantly larger body mass index than the others, so in order to maintain a balance between anthropometric measurements of the participants, they were excluded from the study.

### 2.2. Measurements

The Department of Pharmacology and Therapeutics, Medicine Faculty, Collegium Medicum in Bydgoszcz provided the anthropometric characteristics. Blood samples of the participants were collected at the clinic before and after a specified period of training. Serum was prepared immediately, frozen at −20 °C, and shipped on dry ice to a central facility, where it was stored at −70 °C until assay. Assays were performed using stringent quality control. Biomarkers were determined with the ELISA method on a BioTek EPOCH Instrument (BioTek, Winooski, VT, USA).

### 2.3. Statistical Analysis

Quantitative results were presented as mean values with standard error of the mean (±SEM) and additional minimum and maximum values. The Shapiro–Wilk test was used to check the compliance of the results distribution with the normal distribution for the results obtained before (time point 0) and after the training season (time point 1). The Student’s test provided the comparison of the results having a normal distribution for the dependent variables. The Wilcoxon test was applied when the variables were not normally distributed. An independent *t*-test or a Mann–Whitney U test was used to compare the participants before and after training with the control group. The *p*-values < 0.05 were considered as statistically significant.

## 3. Results

The baseline characteristics include age, body weight, height and BMI (body mass index) in both control and training groups. Following the World Health Organization interpretation of the BMI values, participants were in healthy body mass range or overweight. It is, however, worth mentioning that BMI does not include muscle mass into consideration, therefore an individual with higher than usual percentage of muscle mass could be considered as overweight according to this measurement. In this research, we did not study the correlation between the BMI value and sex hormone concentration, thus state of being overweight was not an exclusion criterium. The mean age of participants was 18.52 ± 0.25 years in the training group and 21.08 years in the control group. The body mass of participants was 79.69 ± 2.11 (kg) in the training group and 76.85 (kg) in the control group (*p* > 0.05). The average height of a participant was 1.87 ± 0.01 (m) in the training group and 1.81 (m) in the control group (*p* > 0.05). The average BMI (body mass index) of the participants were 22.52 ± 0.4 (kg/m^2^) and 23.43 (kg/m^2^) in the control group (*p* > 0.05) (Table 1).

After the training season, we found that both serum testosterone and SHBG concentration had decreased in participants (Table 2). The mean value of testosterone concentration in the control group was 5.6 ± 0.38 ng/mL. Results showed lower values of serum testosterone concentration in the training group before the season and after the season (6.59 ± 0.35 (ng/mL) vs. 7.12 ± 0.4 (ng/mL)). It was also proved in this research that the basal testosterone serum concentration was higher in males in the training group than in the control group (7.12 ± 0.4 (ng/mL) vs. 5.6 ± 0.38 (ng/mL)) as well as the mean body mass of rowers in the training group compared to the control group (79.69 (kg) vs. 76.85 (kg)). Apparent differences were noticed and testosterone level after training season was higher (7.12 ± 0.4 (ng/mL)) than in the control group (5.6 ± 0.38 (ng/mL)), however, the results seem to be clinically insignificant (*p* = 0.055). As the study also showed, a decrease in SHBG serum concentration (from 39.50 ± 2.48 to 34.27 ± 2.33 (nmol/L)) and increase in estradiol level (from 78.2 ± 11.21 to 83.01 ± 13.21 (pg/mL)) were observed. However, presented results, viz the increase in estradiol after the complete training season were statistically insignificant (*p* = 0.345) (Figure 1). Research demonstrated SHBG had significantly decreased comparing pre- and post-season measures, but the concentration between non-training athletes and training group is invariably lower. Our findings also show that the endurance training caused reduction in SHBG serum level. There were no statistically significant differences between examined groups in NO. Based on many previously published papers there is a strong association with testosterone levels and ApoA1 values. In hypogonadal men, after enhancing testosterone level, in this case one month of supplementation treatment with a dose of 100 mg, the researchers observed a decrease in ApoA1, alongside a decrease in HDL cholesterol and an increase in hepatic lipase activity [16]. Similar results were obtained by Berg et al. [17]. In the experiment, after the average 216 mg testosterone treatment, decrease in total HDL and HDL2 cholesterol as well as in ApoA1 were displayed. Although the atherogenic risk expressed as total cholesterol/HDL cholesterol ratio did not change during treatment, the high testosterone levels were associated with significantly lower values of biomarkers of subclinical atherosclerosis. Our results show that with a decrease in a testosterone serum concentration, a minor increase in ApoA1 was observed (*p* > 0.05). Apparently, the serum concentration of the two substances seem to be inversely proportional. The topic on testosterone and ApoA1 values in training are not sufficiently presented, hence there is still a necessity for more thorough research.

## 4. Discussion

Athletes in professional sport are required to perform high-intensity training volumes in order to improve in an anabolic way. Excessively prolonged and intense exercises, stressful competition or other factors can result in underrecovery, which leads to underperformance and progressive fatigue. This is described as overtraining or overreaching (short-term syndrome with quickly reversible symptoms). These states are probably associated with insufficient metabolic recovery, followingly resulting in a decline in ATP levels and even ending up being catabolic.

Sex hormones through different mechanisms have a mediate and immediate impact on Apo-A1, NO and, ultimately, endothelial regulation. Figure 2 presents suggested mechanisms and correlation between assessed molecules. There is a strong evidence that regular exercise provides beneficial effects on cardiovascular health through endothelial function. Training improves endothelium-dependent vasodilator function, firstly, locally in the active muscle group and secondly, as a systemic response when activation involves a vast mass of muscle. NO is the most important cardiovascular signaling molecule. Its intrinsic vasodilator function is commonly used as a substitutive index of endothelial function [18]. Former studies proved that the capability of NO production by endothelium is reduced with aging. A significant receptor for nitric oxide is soluble guanylyl cyclase (sCG), which is activated by production of NO. The activation results in the conversion of guanosine triphosphate (GTP) to cyclic guanosine monophosphate (cGMP) [19]. Former studies proved that the capability of NO production by endothelium is reduced with aging. Except for vascular smooth muscle relaxation, NO also provides the supply of energy. NO, along with cGMP, are mediators, which secure the demand of ATP, especially in tissues requiring high energy supply [20]. In present studies, a significant impact of training on NO serum concentration was not observed (*p* > 0.05). However, interestingly, the concentration of NO in the non-training group was not considerably increased, showing a certain trend of statistical significance.

Estrogen has been known for inducing vasodilation in the cardiovascular system by increasing NO production through multiple pathways: increasing endothelial and neuronal nitric oxide synthase (respectively, eNOS and nNOS), and raising serum concentration of Apo-A1. Endothelial NO production is conclusively promoted by Apo-A1 and thereby controls vascular tone in a process that requires activation of the ecto-F1-ATPase/P2Y1. Testosterone induces enhancement of NO signaling via increased eNOS expression, but diminishes levels of Apo-A1 (reversely to estrogen). In human blood, between 40 and 65% of circulating testosterone and between 20 and 40% of circulating estradiol is bound to SHBG. Binding to SHBG also prevents bound hormones sfrom diffusing out of the bloodstream, thereby preventing hormones binding to the intracellular androgen or estrogen receptors [21].

One of the methods of training is to perform High Intensity Interval Training (HIIT) which is proven to be effective as a way to increase overall endurance, as well as physical strength. The main goal in HIIT training is to perform at submaximal or maximal efficiency (measured by VO2 parameter) in a short amount of time with intervals between exercises reduced as much as possible [22]. This method of training combines benefits of both aerobic and anaerobic activities. It is, however, necessary to provide food with sufficient nutritional values in order to achieve proper recovery in participants. The decrease in hormonal concentration in our study may suggest that participants did not acquire adequate rest quantity and/or nutritional values needed for recovery which resulted in lower hormonal values than before the study.

The primary findings in the study demonstrated that serum levels of testosterone and SHBG have significantly decreased comparing pre- and post-season measures. It is highly unlikely that these changes could be accounted for by decreased testosterone production. Thereby it is proposed that the diminution in serum testosterone concentration reflected an increased demand and utilization of testosterone after the season of intensive training. Similar studies performed by Crewther et al. [23] presented that, collectively, with a reduction of testosterone levels there was an increase in serum cortisol concentration. These findings legitimize the view that catabolic activity is increased during the period after intensive aerobic training. Research suggests that this may be caused by proximate glucocorticoid suppression of steroidogenesis at the Leydig cells [24], inhibition of luteinizing hormone (LH) and gonadotropin-releasing hormone (GnRH) or inhibition of testicular luteinizing hormone receptors [25].

Presented changes in hormone concentration might be a result of long-term sub-maximum endurance training leading to loss of protein available for muscle recovery. As the study also showed a decrease in both serum SHBG and estradiol concentration, we could assess the change in hormonal values as a result of protein loss, and not hormonal imbalances caused by training sessions. This would be possible due to an increase in post-exercise liver blood leading to an increase in testosterone metabolism rate and clearance, with the activation of the parasympathetic nervous system and an increase of aromatization and conversion to estradiol as the other possible outcome [26]. The increase in SHBG levels with 6 months of physical, professional training is probably caused by the induction of protein synthesis in the liver [26,27]. Additionally, recently, muscle strength performance has been positively correlated to SHBG concentration [28]. These findings would be in accordance with our results since we obtained SHBG increase along with higher overall performance of athletes at the end of the training season.

Another possible explanation of this phenomenon would be an increase of androgen receptors as the endurance training performed by the participants is linked to changes in hormonal pathways [29]. Long-term physical activity, an endurance training during the training season in this case, is proved to increase the number of androgen receptors (AR) in the human body. Sex, training status and androgen concentrations are the factors determining the number of the ARs in skeletal muscles [30]. These steroid receptors can mediate the hormonal activity twofold depending on exercise and training. Firstly, by up-regulating (increase in receptor content and/ or hormone binding sensitivity); secondly, by down-regulating (decrease in receptor content and/or hormone binding sensitivity) [31]. The increase of AR allows the body to enhance the uptake of testosterone in tissues in order to increase the protein synthesis and muscle repair, and the process is regulated by the negative feedback loop of the pituitary—gonadal pathway.

Furthermore, this may prove the high-intensity nature of endurance training participants were performing, as the loss of protein (such as SHBG) is linked with increased need of replenishment of nutrients in skeletal muscles. Additionally, both SHBG and testosterone serum levels have decreased proving that participants were partaking in training which caused loss of albumins needed to synthesize hormones. Both aforementioned hormones have a common link in the liver. Albumins, as the most abundant proteins in blood, bind to and transport hormones, including testosterone [32]. The decrease of serum concentration in both hormones may be proven to be caused by excessive training leading to a loss of protein in blood, and not by hormonal imbalances. It is, however, necessary to decrease the serum concentration of SHBG in the post-workout training season in order to increase the concentration of free testosterone in the human body to be available for muscle recovery. The decrease observed in this study may be a result of direct protein loss, but also with the increase of free testosterone in circulation with the decrease of binded (with a SHBG) testosterone [33].

Despite the variations in measurement methods and statistical analyses that might contribute to the discrepancies observed between studies, interesting conclusions are to be noticed. Fink et al. found that typical bodybuilding-type training protocols that include moderate to high intensity, high volume, and comparatively short rest periods are generally effective in inducing acute testosterone increase. This positive correlation between resistance training (RT) and secondary testosterone peak could have been predicted. However, acute elevations of testosterone after RT last only for about 60 min and the average peak generally does not exceed 6.50 ng/mL [31].

Hackney et al. demonstrated that during the 18-week training program, the level of testosterone was reduced from baseline levels (0.117 ± 0.027 (ng/mL)) through the experiment. The greatest decline was observed at week 13 (0.067 ± 0.023 (ng/mL)), subsequently returning to the baseline at the end of the experiment. The participants’ performance was overall improved compared to pre-training measurements. These results show similarities to our study, where testosterone serum concentration was reduced. The possible mechanisms are the result of a higher catabolic activity, overtraining and underrecovery stress factors. An important fact is that decrease in testosterone is not always indicative of compromised performance ability potential. The observation has confirmation in the foregoing study as well as in this research. Even though there was a diminution in introduced hormones levels at the end of the training season, the overall physical durability and maximal performance was increased [34].

Recent studies indicated that estradiol has an important role regarding bone mass, libido and body composition [35], but only if there is a simultaneous testosterone level increase. The concurrent increase has to be linked with higher hypothalamic—pituitary stimulation (through gonadotropin-releasing hormone and luteinizing hormone). Bilha et al. explicate that elevated testosterone consequently provides a natural increase in aromatase activity as a mechanism to avoid excessive testosterone, resulting in a proportional increase in estradiol. The same results were observed in our study with reduction of testosterone (*p* < 0.05) and estradiol elevation (*p* = 0.345). During overtraining syndrome (OTS), increased estradiol with a paradoxical decrease of testosterone hypothetically represents an abnormally increased level of aromatase enzyme, which physiologically is unlikely to be reached. Furthermore, a reduced testosterone:estradiol ratio may trigger an undesirable catabolic state. Low testosterone concentration in OTS may be a consequence of increased conversion to estradiol, because gonadotropic stimulation is maintained. Further work is needed to better understand the phenomenon and possible effects of low sex hormone levels after an endurance training season.

In a study performed on overweight sedentary men, after exercise training with increased dietary intake, Apo-A1 was higher [36]. Another research suggested that moderate intensity exercises in elderly people caused the plasma Apo-a1 level to significantly increase [37]. In our study, no significant enhancement in Apo-A1 serum concentration was presented. The dissimilarities arose from different profiles of participants with a variety of training, however, the conclusion seems to be resembling that there is a direct connection between training and Apo-A1 level. Increased level of the molecule can improve the cardiovascular risk profile of overweight men, but with OTS syndrome, when a loss of proteins can be observed and gonadotropic stimulation is maintained, the low testosterone concentration may cause diminished Apo-A1 serum concentration growth.

## 5. Conclusions

In order to compare and contrast the findings with more possible mechanisms, further and more specific research should be performed. While the decrease in serum concentration levels of such hormones as testosterone would seem detrimental to the physical performance in athletes, our study finds possible explanations for such phenomenon. Furthermore, the baseline level of training groups’ testosterone serum concentration after the decrease is still at a significantly higher level compared to the control group, therefore, the benefits of regular physical activity are still undeniable. In order to provide a better in-depth look into the mechanism of testosterone serum levels, it would be beneficial to create a study with more than one fraction of testosterone.

## Figures and Tables

**Figure 1 biology-12-00531-f001:**
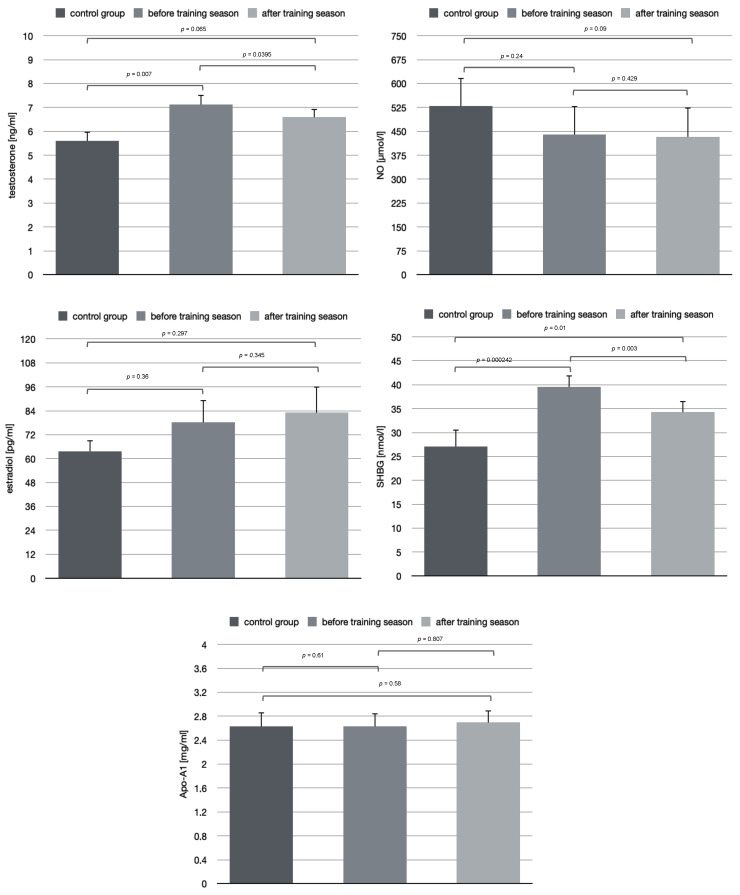
Serum concentration of selected molecules.

**Figure 2 biology-12-00531-f002:**
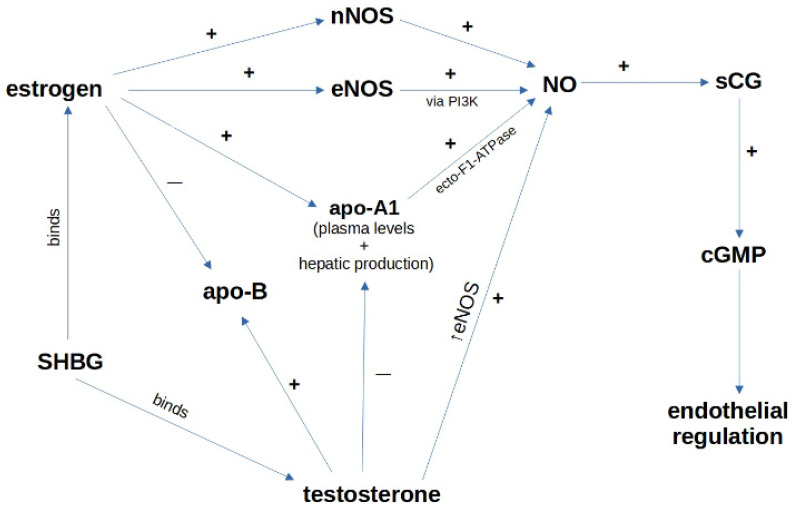
Effect of testosterone on endothelial activity of different molecules.

**Table 1 biology-12-00531-t001:** Anthropometric measurements.

	Mean ± SD		*p*-Value
	Training Group (*n* = 23)	Control Group (*n* = 24)	
Age (years)	18.52 ± 0.25	21.08 ± 0.40	<0.001
Body Mass (kg)	79.69 ± 2.21	76.85 ± 2.26	0.36
Height (m)	1.87 ± 0.01	1.81 ± 0.01	<0.001
BMI (kg/m^2^)	2.52 ± 0.40	23.43 ± 0.62	0.23

**Table 2 biology-12-00531-t002:** Mean values of testosterone, SHBG, Estradiol, NO and APO-A1 serum concentrations in training groups before and after the training season and in the control group.

	Training Group		Control Group
	T_0_	T_1_	
Testosterone (ng/mL)	7.12 ± 0.40	6.59 ± 0.35 *^, △^	5.60 ± 0.38 ^φ^
SHBG (nmol/mL)	39.50 ± 2.48	34.27 ± 2.33 *^, △^	27.08 ± 3.50 ^φ^
Estradiol (pg/mL)	72.20 ± 11.21	83.01 ± 13.20	63.67 ± 5.46
NO (μmol/L)	440.21 ± 88.64	432.75 ± 91.89	528.61 ± 89.32
APO-A1 (mg/mL)	2.62 ± 1.06	2.69 ± 0.98	2.62 ± 0.23

* *p* < 0.05—before training group vs. after training group. ^φ^
*p* < 0.05—control group vs. before training group. ^△^
*p* < 0.05—control group vs. after training group.

## Data Availability

The data presented in this study are available upon request from the corresponding author.

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
