# Peer review of "Does Intense Endurance Workout Have an Impact on Serum Levels of Sex Hormones in Males?"

_biology, 2023, doi:10.3390/biology12040531_

Round 1
Reviewer 1 Report
The authors investigated the sex hormones alternations after rowing training. The results showed that exercise training decreases serum testosterone, sex hormone-binding globulin, and nitric oxide concentrations. This manuscript is fascinating, but some texts are unnecessary (e.g., the sentence about doping and testosterone structure).
Please reconsideration as the following points.
Introduction
As authors know, numerous studies demonstrated that strenuous physical activities alter sex hormones such as testosterone. The introduction was very detailed and written about testosterone. However, I didn't find the aim of this study or the hypothesis. So please clarify what the authors are curious to know. Additionally, why do authors measure Apo-A-I?
Method
When obtained the control blood sample? T0 or T1?
In addition, please add the condition of blood sampling (e.g., overnight fasting before or after the exercise.).
Results
Please add the SEM to the control group's age, body mass, height, and BMI. Does the age of the training group may higher the control? Please add the p-value.
According to the former comment, I can't read what authors want to know, and it is difficult to review, including discussion. However, after the revision, I will check again.
Author Response
Manuscript biology-2247328
Dear Assistant Editor Abby Shi,
Thank you for giving us the opportunity to submit a revised draft of the manuscript “Does intense endurance workout have an impact on serum levels of sex hormones in males?” for publication in the Biology Journal. We appreciate the time and effort that you and the reviewers dedicated to providing feedback on our manuscript and are grateful for the insightful comments on and valuable improvements to our paper. We have incorporated most of the suggestions made by the reviewers. Please see below for a point-by-point response to the reviewers’ comments and concerns. All page numbers refer to the revised manuscript file with tracked changes.
Sincerely,
Oskar Kuźmiński
Department of Pharmacology and Therapeutics
Reviewers’ Comments to the Authors:
- This manuscript is fascinating, but some texts are unnecessary (e.g., the sentence about doping and testosterone structure).
Authors response: Thank you. After reconsideration we found those two texts unnecessary.
- As authors know, numerous studies demonstrated that strenuous physical activities alter sex hormones such as testosterone. The introduction was very detailed and written about testosterone. However, I didn't find the aim of this study or the hypothesis. So please clarify what the authors are curious to know.
Authors response: Thank you for pointing this out. The reviewer is correct, and we have added the essence.
„The aim of this study was to investigate the relationship between endurance training and concentrations of testosterone, SHBG, estradiol, NO and Apo-A1 to suggest a potential mechanisms for changes in sex hormones after endurance training/workout.„ page 3
3. Additionally, why do authors measure Apo-A-I?
Authors response: We agree with the reviewer’s assessment. Accordingly we have added a part about the Apo-A1 measures.
„Based on many previously publicised papers there is a strong association with testosterone levels and ApoA1 values. In hypogonadal men after enhancing testosterone level, in this case one month of supplementation treatment with a dose of 100mg, the researchers observed a decrease in ApoA1, alongside with decrease in HDL cholesterol and an increase in hepatic lipase activity. [38]. Similar results were obtained by Berg et al. [39]. In the experiment after the average 216 mg testosterone treatment decrease in total HDL and HDL2 cholesterol as well as in ApoA1 were showed. Although the atherogenic risk expressed as total cholesterol/HDL cholesterol ratio did not change during treatment, the high testosterone levels were associated with significantly lower values of biomarkers of subclinical atherosclerosis. Our results show that with a decrease in a testosterone serum concentration a minor increase in ApoA1 was observed (p>0.05). Apparently the serum concentration of the two substances seem to be inversely proportional. The topic on testosterone and ApoA1 values in training are not sufficiently presented, hence there is still a necessity for more thorough research.” page 5
4. When obtained the control blood sample? T0 or T1? In addition, please add the condition of blood sampling (e.g., overnight fasting before or after the exercise.)
Authors response: Thank you for pointing this out. The details has been added in writing.
„The condition of blood sampling was identical in both T0 (before the training season) and T1 (after the training season) - the participants were asked to remain in a fasted state (with the last meal scheduled to be in the evening the day before the blood collecting). The samples were drawn between 9AM and 10AM in both T0 and T1 in a laboratory environment.” page 3
5. Please add the SEM to the control group's age, body mass, height, and BMI. Does the age of the training group may higher the control? Please add the p-value.
Authors comment: We think this is an excellent suggestion. The SEM and p-values was supplemented in the Table 1, as recommended. We have made some insights about that case. There is a difference between the age of the control and training group. We think that despite the statistical dissimilarity, a minor age difference is not clinically significant and does not practically influence the values of the assessed molecules- testosterone, NO, estradiol, SHBG and Apo-A1.
Table 1. Anthropometric measurements, page 6
6. This is a very intersting and important topics but you do not present it well enough for giving the real insight of your research. Furthermore, you do not give any hypothesis at the end of your introduction and this article seems to be a kind of review including the authors results without any discussion about the point.
Authors response: Thank you. While we appreciate the reviewer’s feedback, we respectfully disagree. We think that the individual molecules were discussed, confronted and compared to the previous research that were made on that ground. Each point considered significant from the clinical point of view is presented. Our insights about possible explanations for such a results were also introduced.

Reviewer 2 Report
Your abstract give the main results but you DO NOT discuss them and you article is presented as a teaching lesson on the topics.
Furthermore, you do not give any hypothesis at the end of your introduction and this article seems to be a kind of review including the authors results without any discussion about the point
This is a very intersting and important topics but you do not present it well enough for giving the real insight of your research.
Please try to present your article as a real experimental original research and include more references about avertraining and hormone in endurance sport for the discussion of each points of your results:
The direct effect of physical activity was decrease in testosterone serum concentration (from
7.12 ±0.4 to 6.59 ±0.35 [ng/mL]), sex hormone binding globulin serum concentration (from 39.50
±2.48 to 34.27 ±2.33 [nmol/L]), nitric oxide serum concentration (from 440.21 ±88.64 to 432 ±
0.35 [ng/mL]), increase in estradiol serum concentration (from 78.2 ±11.21 to 83.01 ±13.21
[pg/mL]) and no significant increase in Apo-A1 serum concentration (from 2.63 ±0.2 to 2.69 ±0.21
[mg/mL]). Authors analysed the data and reviewed the former conterminous articles to find the
possible mechanisms leading to changes of serum concentration of certain hormones and molecules.
Author Response

(The authors gave the same response as above.)

Round 2
Reviewer 1 Report
According to the reviewer’s comments, the authors revised appropriately.
The authors revised the introduction about testosterone. Therefore, I suggest the sentence “ 1.1 Correlation between level of testosterone and physical performance.” combine the above sentence, which describes testosterone (Testosterone retains nitrogen and …).
Author Response
Thank you! We revised the manuscript with the latest suggestion. The revised manuscript is submitted.